# Acceptance of a Mobile Telepresence Robot, before Use, to Remotely Supervise Older Adults’ Adapted Physical Activity

**DOI:** 10.3390/ijerph20043012

**Published:** 2023-02-09

**Authors:** Nicolas Mascret, Jean-Jacques Temprado

**Affiliations:** Aix Marseille Univ, CNRS, ISM, Marseille, France

**Keywords:** telepresence, health, expectations for aging, technology acceptance model, robot

## Abstract

Many older adults remain sedentary because they do not have access to specialized facilities or adapted physical activity (APA) teachers. To solve this health issue, mobile telepresence robots (MTRs) could allow APA sessions to be supervised by a teacher from a distant location. However, their acceptance has never been investigated in the context of APA. A sample of 230 French older adults filled out a questionnaire assessing the variables of the Technology Acceptance Model and their expectations for aging. The results showed that the more the older adults found the MTR useful, easy to use, enjoyable, and recommended by their entourage, the more they intended to use it. Moreover, older adults who expected higher health-related quality of life with aging are those who found the MTR more useful. Finally, older adults significantly found the MTR useful, easy to use, and pleasant to use for remote supervision of their physical activity.

## 1. Introduction

### 1.1. Background

For a growing number of older adults, preventing the deleterious effects of aging on mental health and functional capacities is of major importance [1]. Research has repeatedly shown the benefits of physical activity in this respect (for a review, see [2]). Unfortunately, however, due to their social isolation, many older adults do not have access to facilities or specialized teachers to practice supervised physical activity regularly [3,4].

The COVID crisis has demonstrated that new technologies, in particular, static videoconference (e.g., [5]), have an important role to play in overcoming this problem [6]. However, static videoconference limits the displacements of the APA teacher around the participants, thereby making it difficult to deliver accurate feedback, which is critical for training effectiveness. Moreover, during static videoconference, the restricted field of the camera obliges older adults to limit their movements and displacements to be seen continuously by the APA teacher. Consequently, several motor tasks cannot be used, which reduces the effectiveness of static videoconference. On the other hand, because they do not have these kinds of limitations, mobile telepresence robots (MTRs) seem more promising for remotely supervising older adults’ physical activity. Many different models of MTRs are available on the silver economy market (e.g., Sam, Buddy, Ubbo, Robear, IPal), which have in common that they are equipped with an interactive two-way video and audio on a mobile embodiment. An MTR allows the user facing the robot to have the impression that the operator (i.e., the APA teacher) is present, though he/she is distant [7]. The combination of videoconference and the robot’s mobility allows the operator to see, hear, and move throughout the (fitness) room in which the participants perform the requested exercises [8].

Despite their differences, both static and mobile videoconference allow the provision of supervised physical activity programs, which have been shown to be more effective than unsupervised ones [5]. However, whether supervising physical activity programs through an MTR could be more effective than doing it through static videoconference is a matter of debate. We contend that, thanks to the technical sophistication of most MTRs (e.g., mobility, screen rotation), they could allow the above-mentioned limitations of fixed videoconference to be circumvented, while offering extended possibilities for the APA teacher to intervene effectively (e.g., to observe participants’ motor activity to provide more accurate feedback), as if he/she were present in the room. Finally, because the MTRs are piloted by a specialized APA teacher, this should make it possible to better adapt to older adults, which is a problem often encountered with other technological solutions (e.g., autonomous robots, [9]).

However, though one can be confident about the effectiveness of MTRs, the question of their acceptance by older adults remains open. The main purpose of the present study was to address this issue.

### 1.2. Technology Acceptance and MTR Acceptance

Studies on technology acceptance examine the psychological determinants of the intention to use a technology. This may be investigated after a long effective use of a technology, but also before its use (i.e., without any experience of it), which is sometimes called “acceptability” [10]. In the present study, we will use the term “acceptance before use”, which is by far the most used terminology in the literature. Investigating acceptance of a technology before a first use is of great interest for several reasons [11]: (i) even if individuals have never used a technology, they may have an opinion on it, (ii) acceptance is a first step before the roll-out of a technology, and (iii) it allows the identification of some psychological antecedents that could promote, block, or threaten the intention to use it.

Since the late 1980s, several models have been developed to study technology acceptance (before use). The most widespread model is the Technology Acceptance Model (TAM, [12,13,14]), which includes two main component variables: perceived usefulness (i.e., the degree to which an individual believes that using a system could improve his/her performance) and perceived ease of use (i.e., the degree to which an individual believes that using the system could be effortless). These two variables are positive predictors of intention to use the system: the more useful and easier to use an individual perceives a technological device to be, the more likely it is that he/she has the intention to use it. Naturally, effective use of the system was not retained as a variable of interest in the present study since we were investigating technology acceptance before first use. However, we complemented the TAM with two other variables: perceived enjoyment (i.e., the degree to which the use of technology is perceived to be enjoyable, regardless of its effectiveness) and subjective norms (i.e., the degree to which an individual perceived that the people around him/her would encourage him/her (or not) to use this technology). In the TAM literature [15,16], perceived enjoyment is currently considered a positive predictor of intention to use a technology, while subjective norms were positive predictors of perceived usefulness, perceived ease of use, and perceived enjoyment. 

Several studies have specifically investigated the acceptance of MTRs by older adults to attenuate social isolation and loneliness (for a review, see [17]). Results showed that older adults were satisfied with the appearance of an MTR, which was found to be convenient to use, fun, and they were also happy to interact with it. An MTR was also perceived as useful because it facilitated communication with their family, while being more attractive and interactive than a phone call [17]. MTRs enabled effective communication, transmitting between the two interlocutors sound, image, gestures, physical movements, but also environmental information [18]. Moreover, another study has shown that MTRs are well-accepted by older adults in the healthcare domains, such as medical teleconsultation through MTR [19]. Strikingly, only one study has investigated MTR acceptance for remotely supervising older adults’ physical activity [20]. Results showed that, before a first use, MTRs’ perceived usefulness and perceived ease of use were quite high for community-dwelling older adults with mobility impairment. Though these results were encouraging with respect to the acceptance of MTRs by older adults to remotely supervise their physical activity, this study also had some limitations: (i) it focused on a clinical population of older adults (i.e., community-dwelling older adults with mobility impairment), making it difficult to extend its findings to a population of healthy and active older adults, which is targeted in the present study; (ii) the sample size was quite small (14 participants), (iii) no statistical analyses were conducted on the data to confirm that MTRs’ perceived usefulness and perceived ease of use were significantly higher than the mean of the scale, and (iv) the TAM was not validated as a model: only two of the TAM variables were used, which limited the scope of the results of this kind of studies compared with those that fully validate the model by investigating the relationships among the variables [21]. Using the TAM and a large sample of older adults, the aim of the present study was to investigate the acceptance of the MTR before its first use by older adults to remotely supervise their physical activity. Moreover, to increase the explanatory power of the TAM, we added an external variable, namely expectations for aging.

Expectations for aging are of interest since they can modulate the acceptance of MTRs. When they are high, this indicates that older adults expect achievement and maintenance of high physical and mental functioning with aging; while, when expectations for aging are low, this indicates that older adults expect decline with aging [22]. The role of perceptions of aging in older adults’ future health outcomes has been previously demonstrated [23]. For instance, older adults with high expectations for aging were more likely to report a high level of physical activity than those with lower age expectations, even after controlling many other variables (e.g., age, gender, level of education, comorbidity). These results have been confirmed with older adults of low socioeconomic status [24]. It is worth noting that low expectations for aging are more detrimental to the adoption of appropriate health behaviors (e.g., physical activity) than the benefits of having high expectations for aging [25]. Thus, assessing expectations for aging may be of interest because it is related to health benefits resulting from regular physical activity made possible using MTRs [2]. To the best of our knowledge, expectations for aging had never been used in any study focusing on technology acceptance by older adults, regardless of technology. The present study was a step in this direction. 

### 1.3. Objectives and Hypotheses of the Present Study

The first aim of the present study was to test the validity of the TAM, including expectations regarding aging as an external variable, to assess the acceptance of an MTR used to remotely supervise older adults’ physical activity. Based on the available literature, the following hypotheses were tested, i.e., that intention to use the MTR by older adults for their physical activity would be positively predicted by perceived usefulness, perceived enjoyment, perceived ease of use, and subjective norms. In addition, we hypothesized that perceived usefulness of the MTR would be positively predicted by expectations regarding aging. Specifically, we predicted that older adults with high age expectations would find the MTR useful to remotely supervise their physical activity, while older adults with low age expectations would find it useless. 

Ancillary hypotheses may also be formulated: perceived usefulness would be positively predicted by perceived enjoyment, perceived ease of use, and subjective norms. Perceived enjoyment would be positively predicted by perceived ease of use and subjective norms. Finally, perceived ease of use would be positively predicted by subjective norms. All these hypotheses are presented in Figure 1.

The second aim of the present study was to investigate the level of acceptance of the MTR by older adults for remotely supervising their physical activity. We hypothesized that perceived usefulness, perceived ease of use, perceived enjoyment, subjective norms, and intention to use the MTR would be significantly higher than the mean of the Likert scale, indicating that the MTR would be generally well accepted by older adults before a first use.

## 2. Methods

### 2.1. Participants and Procedure

A sample of 230 French older adults (154 women, 76 men) aged between 55 and 84 (*M*_age_ = 66.61 years, *SD* = 7.06) participated in the study. Two eligibility criteria were used in the present study: (1) aged 55 and above, and (2) living in a private home without daily assistance with everyday living tasks (e.g., grooming, meal preparation), or only household help once or twice a week. The age of 55 is frequently used as the threshold for determining who may be considered an older adult (e.g., [26]). 

The data collection was completed in individual paper-based sessions. First, the participants read a brief text explaining what an MTR is, its functionalities, and its potential use to remotely supervise older adults’ physical activity. Photos of the MTR accompanied the text because many older adults had never seen an MTR before. This procedure is classically used in studies conducted with the TAM before the first use of the technological device, especially in older adults (e.g., [27]). Then, the participants anonymously filled out the questionnaire containing the focal constructs (TAM variables, expectations for aging) and demographic information (gender, age, level of education, financial status). Participants could stop completing the anonymous questionnaire at any time. The present study has been approved by the National Ethics Committee (IRB00012476-2021-10-03-93).

### 2.2. Measures

Participants responded to the four items assessing perceived usefulness (e.g., *“Using this robot would allow me to do physical activity efficiently”*), the three items assessing perceived ease of use (e.g., *“I think it would be easy to learn how to use this robot”*), the three items assessing perceived enjoyment (e.g., *“Using this robot would be fun”*), the three items assessing subjective norms (e.g., *“The people who are important to me would encourage me to use this robot”*), and the three items assessing intention to use (e.g., *“If I had the opportunity to have easy access to this robot, I would like to use it”*) on a Likert scale from 1 (*strongly disagree*) to 5 (*strongly agree*). These items were developed based on items commonly used in the TAM and on other studies specifically investigating technology acceptance by older adults [15,28]. 

Expectations regarding aging were assessed by the short version of the Expectations Regarding Aging Survey (ERA-12, [22]). The participants responded to the twelve items (e.g., *“When people get older, they need to lower their expectations of how healthy they can be”*) on a 4-point Likert scale (from *“Completely wrong”* to *“Completely right”*). The present study used the total score of the twelve items following the calculation procedure [22]. A higher score means that older adults expect higher health-related quality of life with aging, and a lower score means that they expect decline.

Internal consistency was acceptable for each variable, with McDonald’s omegas [29] ranging from 0.88 to 0.96. Descriptive statistics, normality of the distribution, and internal consistency are presented for each variable in Table 1.

### 2.3. Statistical Analyses

All the data analyses were conducted using the JASP software (version 0.16.2) with a level of significance defined at *p* < 0.05. Preliminary data analyses were performed to detect gross outliers using the Mahalanobis distance (*χ*^2^(9) = 27.88, *p* < 0.001) at the multivariate level [30]. Moreover, variables non-normal in distribution were identified with values ≥ |2| for skewness and ≥ |7| for kurtosis [31].

Following the recommendations [32,33], the model fit was evaluated using several fit indices: the *χ*^2^*/df* ratio (a value ≤ 3 is needed), the CFI (Comparative Fit Index; a value ≥ 0.90 is needed), the TLI (Tucker–Lewis Index; a value ≥ 0.90 is needed), the RMSEA (Root Mean Square Error of Approximation; a value ≤ 0.08 is needed), and the SRMR (Standardized Root Mean Square Residual; a value ≤ 0.08 is needed).

A structural equation modeling (SEM) was then performed with the maximum likelihood estimation [34] to test the hypotheses provided in Figure 1. Gender, age, level of education, and financial status were also entered in the model to control for these variables, but the search for parsimony led us to remove a control variable if it was not identified as a significant predictor of the model variables [35].

Potential differences between the mean of the scale (i.e., 3) and the scores of the TAM variables (perceived usefulness, perceived ease of use, perceived enjoyment, subjective norms, and intention to use) were examined for the whole sample using five consecutive one-sample *t*-tests (one per variable). Cohen’s *d* was used to calculate effect sizes.

## 3. Results

### 3.1. Preliminary Results

Among the participants, 26.1% of the participants were aged between 55 and 64, 54.3% between 65 and 74, and 19.6% over 75. Participants’ level of education was self-reported: no qualification (0.9%), qualification below baccalauréat (10.9%), intermediate vocational qualification (13.9%), baccalauréat (15.2%), 2 years of HE (21.7%), 3+ years of HE (37.4%). Four categories of financial status were identified: adequate financial resources (46.1%), adequate financial resources, except for unforeseen circumstances (32.2%), limited financial resources (15.6%), lack of financial resources (6.1%).

Using the Mahalanobis distance, one participant was identified as an outlier and was removed from the final sample. Based on the values of skewness (max = |0.62|) and kurtosis (max = |1.28|) presented in Table 1, normality assumptions were validated.

### 3.2. Validation of the TAM

First, a confirmatory factor analysis (CFA) was conducted on the covariance matrix of the TAM items, and the results supported the hypothesized six-factor model (*χ*^2^(335, *N* = 230) = 526.71, *p* < 0.001, *χ*^2^*/df* = 1.57, CFI = 0.99, TLI = 0.99, RMSEA = 0.05, SRMR = 0.06). The results of the SEM analysis showed that the model fit met the expected requirements (*χ*^2^(5, *N* = 230) = 11.20, *p* = 0.05, *χ*^2^*/df* = 2.24, CFI = 0.99, TLI = 0.98, RMSEA = 0.07, SRMR = 0.04). Then, the main results of the SEM showed that older adults’ intention to use the MTR for their own physical activity was positively predicted by perceived usefulness (*p* < 0.001), perceived enjoyment (*p* < 0.001), perceived ease of use (*p* = 0.04), and subjective norms (*p* = 0.01). In other words, the more the older adults found the MTR useful, easy to use, enjoyable, and recommended by their entourage, the more they intended to use it. The model predicted 84.4% of the variance of intention to use. The main results of the SEM also highlighted that expectations regarding aging positively predicted perceived usefulness (*p* < 0.001), indicating that older adults who expected higher health-related quality of life with aging are those who found the MTR more useful. 

Ancillary results showed that perceived usefulness was positively predicted by perceived enjoyment (*p* < 0.001), perceived ease of use (*p* = 0.001), and subjective norms (*p* < 0.001). Perceived enjoyment was positively predicted by perceived ease of use (*p* < 0.001), and subjective norms (*p* < 0.001). Finally, perceived ease of use was positively predicted by subjective norms (*p* < 0.001). Regarding the control variables (gender, age, level of education, financial status), only level of education negatively predicted perceived enjoyment (*β =* −0.100, *p* = 0.04) and positively predicted perceived ease of use (*β =* 0.190, *p* < 0.001). For the sake of clarity, the results of this control variable do not appear in Figure 2, which represents the final model that was validated.

### 3.3. Levels of MTR Acceptance 

The results of the one-sample *t*-tests highlighted that the means of perceived usefulness (*t*(229) = 2.40, *p* = 0.02, *d* = 0.16), perceived ease of use (*t*(229) = 7.48, *p* < 0.001, *d* = 0.49), and perceived enjoyment (*t*(229) = 2.32, *p* = 0.02, *d* = 0.15) were significantly higher than the mean of the Likert scale (i.e., 3), indicating that older adults found the MTR more useful, easier to use, and more pleasant to use than the mean, for supervising remote physical activity. However, no significant differences were identified for subjective norms (*p* = 0.41) and intention to use (*p* = 0.65). These last results indicated that, before a first use, (i) the older adults thought that those around them would not encourage them to use the MTR for physical activity, but would not discourage them either, and (ii) the older adults had overall neither the intention to use the MTR nor the intention not to use it. The results are presented in Figure 3. 

## 4. Discussion

The results of the present study suggested that perceived usefulness, perceived enjoyment, perceived ease of use, and subjective norms were positive predictors of older adults’ intention to use the MTR for practicing physical activity under the supervision of a distant PA teacher. In other words, the more the older adults found the MTR useful, easy to use, enjoyable, and recommended by their entourage, the more they intended to use it. These findings validated the TAM for: (i) a specific technology (MTR), (ii) an original purpose (remotely supervising physical activity), and (iii) a particular population (older adults). 

The other analyses showed that the means of perceived usefulness, perceived ease of use, and perceived enjoyment were significantly higher than the mean of the Likert scale, indicating that older adults found the MTR more useful, easy to use, and pleasant to use than the mean for supervising remote physical activity. However, no significant differences were identified for subjective norms and intention to use. 

The main predictor of intention to use the MTR was its perceived usefulness, which was also significantly higher than the mean of the scale. This is not surprising, since perceived usefulness is usually and conceptually the strongest predictor in the TAM literature [12]. These results are important to take into consideration because perceived usefulness is the crucial variable for pre-implementation acceptance of technology by older adults, i.e., when a device has not yet been tried [36]. Importantly, it leads one to be confident of the acceptance of the MTR for supervising remote physical activity in older adults. These results are consistent with previous studies showing that perceived usefulness is a major variable to better understand acceptance of utilitarian technologies [37]. MTRs might be considered by older adults a utilitarian technology because they would allow them to benefit from remote supervision of their physical activity. Notably, however, the intention to use MTRs was not significantly higher than the mean of the scale (without being lower either), indicating that the older adults have neither the intention to use the MTR nor the intention not to use it. Because this study focused on acceptance of the MTR prior to actual first use, it might be that the older adults were waiting to use it before positioning themselves on their intention to use it. It might explain why their scaling was around the mean.

It is worth noting that perceived enjoyment was found to be a strong predictor of older adults’ intention to use the MTR. Perceived enjoyment was also higher than the mean of the scale. Perceived enjoyment was integrated in the original TAM and is currently considered a major factor motivating the use of technologies [38]. This was confirmed by the results of the present study, which could appear surprising at first sight. One might have thought that the level of perceived enjoyment would not be so high because some form of anxiety or fear about the MTR might emerge [39]. Although anxiety about robots (in general) and about MTRs (in particular) was not directly measured in this study, the high level of perceived enjoyment related to the use of the MTR suggests that anxiety does not appear to be a factor negatively impacting the intention to use the MTR. However, this hypothesis needs to be confirmed using the appropriate scales assessing robot anxiety. 

Like perceived usefulness and perceived enjoyment, perceived ease of use positively predicted older adults’ intention to use the MTR and was higher than the mean of the scale. Perceived ease of use was the lowest predictor of intention to use, while remaining a significant predictor, indicating that this predictor is relevant for older adults who are not always familiar with technologies [27] but also that it is not the most important factor for them in terms of MTR acceptance. This pattern of results has been found in several studies conducted with older adults and with other innovative technologies such as virtual reality head-mounted display [15]. Perceived ease of use was also found in the present study to be the variable with the highest score compared with the mean of the scale. This can be explained by the fact that the older adults understood that the MTR was to be remotely piloted by the APA teacher and that they therefore had little intervention to perform themselves. 

Subjective norms were found to be a positive predictor of perceived usefulness, perceived ease of use, perceived enjoyment, and intention to use the MTR, indicating that the more the older adults perceived that the people around them would encourage them to use the MTR, the more they found it useful, easy to use, and enjoyable, and the more they had the intention to use it for supervising their physical activity. These results can be explained by the fact that older adults with a higher score on subjective norms thought that they received better support from their family, their friends, and their peers [40], who would like them to be able to benefit from remote physical activity supervision, which is known to have health benefits for older adults. This explanation seems valid for the older adults with the highest scores on subjective norms, but it is less appropriate for the whole study population because no significant differences from the mean of the scale have been identified for subjective norms. In other words, the older adults thought that those around them would not encourage them to use the MTR for physical activity but would not discourage them either, perhaps because the MTR is an unusual and not yet widespread technology.

Finally, according to our hypothesis, expectations regarding aging positively predicted perceived usefulness, indicating that older adults who expected higher health-related quality of life with aging are those who found the MTR more useful. This result was not surprising, since high perceptions of aging influence many older adults’ future health outcomes [22], such as engaging in physical activity [23,24], which is of particular interest for preventing the deleterious effects of aging [2]. This might explain why older adults with high expectations for aging have found the MTR useful for supervising their physical activity, highlighting for the first time, regardless of the technology, that expectations for aging influenced older adults’ technology acceptance.

### Current Limitations and Directions for Future Studies

First, the participants filled out self-reported questionnaires, which may induce social desirability [41] since older adults do not necessarily want to admit that they are not very comfortable with technologies, especially those as original and unusual as the MTR. However, this risk remained under control because the completion of the questionnaires was anonymous in the present study and the results showed that some of the TAM variables (perceived usefulness, perceived ease of use, perceived enjoyment) were significantly higher than the mean of the scale, without reaching high scores, while other variables were not identified as significantly different from the mean of the scale (subjective norms and intention to use). If a social desirability phenomenon had occurred, all the TAM variables would have scored significantly higher than the scale mean, with very high scores between 4 and 5. This was not the case in the present study. Secondly, one of the usual criticisms of the TAM is that its explanatory power remains rather limited (between 30 and 40%), which has led some authors to propose another model, the Unified Theory of Acceptance and Use of Technology (UTAUT, [42]), to increase the percentage of variance explained (between 60 and 70%) by integrating other predictors of the intention to use. However, the explanatory power of the tested TAM in the present study was very high since the final model predicted 84.4% of the variance of intention to use. Thirdly, although the present study focusing on MTR acceptance for supervising physical activity provided interesting results before a first use of this technological device, it does not inform as to its acceptance after an actual use. The different variables assessing technology acceptance may increase or decrease after a first use or more regular use [13]. A future study using pre-test and post-test measures of acceptance might be relevant to investigate how older adults’ acceptance of the MTR would evolve after a physical activity session supervised by an APA teacher through the MTR. To go one step further, it may also be relevant to examine how MTR would be adapted to everyday life and how it would contribute to changes in everyday life, based on the domestication paradigm [43]. Fourthly, although they are the ultimate recipients, the potential users of MTR are not limited to older adults. APA teachers are also potential users, as pilots of the robot. Therefore, it would be interesting to measure both their acceptance of this device before its first use (to identify possible blockages) and after its actual use. If older adults accept the MTR for supervising physical activity (as identified in the present study) but APA teachers do not, there is a risk that the MTR will never be used.

## 5. Conclusions

While regular practice of physical activity programs is highly recommended in older adults [4], many older adults remain distant from APA structures and teachers, thereby potentially losing chances of aging well. Thus, MTRs appear to be a promising solution to overcome this obstacle by allowing remote supervision of physical activity in older adults. However, its acceptance by older adults needed to be measured before it could be used for the first time. Based on a questionnaire study, this was the aim of the present study, which has validated the TAM applied to this device (MTR), this purpose (remote physical activity), and this population (older adults), including an external variable in the model, namely expectations regarding aging. The level of acceptance of the MTR in the present study was found to be quite high, especially for perceived usefulness, perceived ease of use, and perceived enjoyment. In other words, no obstacle was identified among older adults regarding the acceptance of this device for remote supervision of physical activity. From now on, its effectiveness for supervising APA sessions must be validated, as well as its acceptance by older adults after an effective first use and its acceptance before and after use by its other potential users, namely the APA teachers. Under these conditions, the present MTR appears to be a potential solution to allow the remote supervision of physical activity by professionals for older adults who do not have access to APA structures or teachers.

## Figures and Tables

**Figure 1 ijerph-20-03012-f001:**
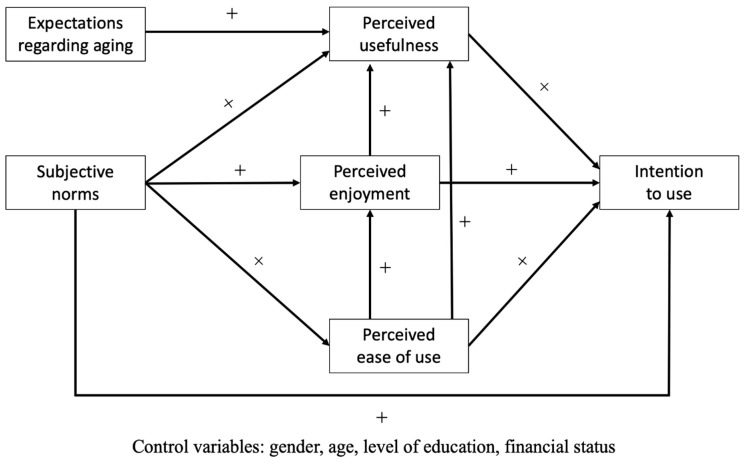
The hypothesized model. Note: + indicates a positive prediction.

**Figure 2 ijerph-20-03012-f002:**
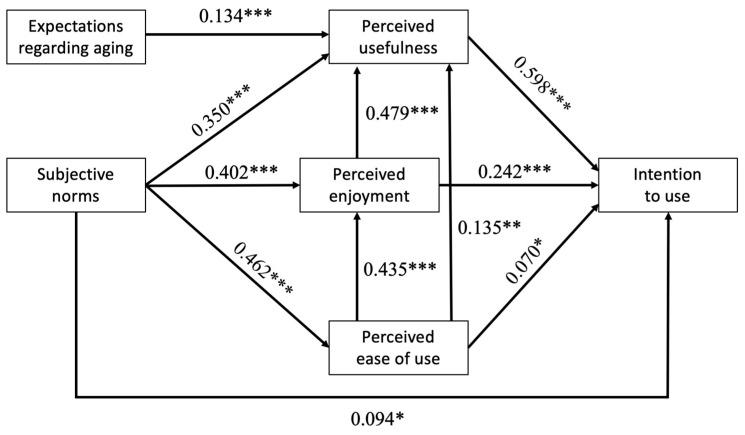
Validated structural model with standardized path coefficients. * *p* < 0.05, ** *p* < 0.01, *** *p* < 0.001.

**Figure 3 ijerph-20-03012-f003:**
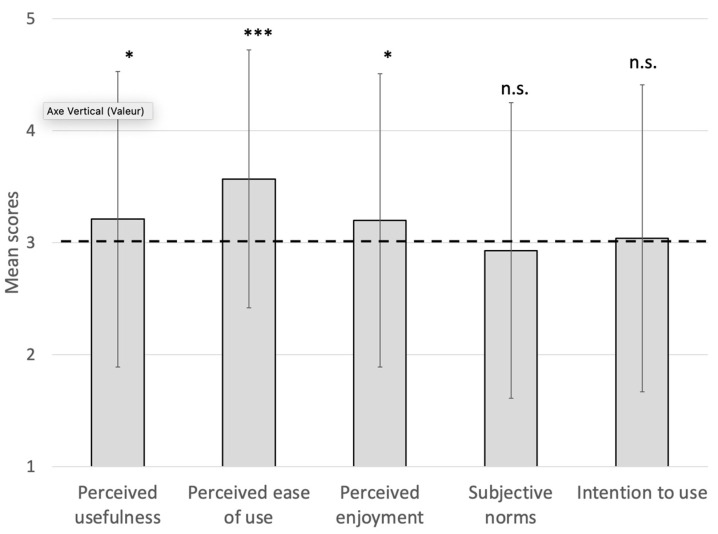
Comparison with the mean of the scale for each variable of the TAM. The dotted line represents the mean of the scale. * *p* < 0.05, *** *p* < 0.001, n.s. = non-significant.

**Table 1 ijerph-20-03012-t001:** Descriptive statistics, skewness, kurtosis, and internal consistency.

	Perceived Usefulness	Perceived Ease of Use	Perceived Enjoyment	Subjective Norms	Intention to Use	ERA
Mean	3.21	3.57	3.20	2.93	3.04	51.40
Standard deviation	1.32	1.15	1.31	1.32	1.37	17.60
Skewness	−0.27	−0.62	−0.22	−0.24	−0.08	−0.38
Kurtosis	−1.14	−0.50	−1.06	−1.09	−1.28	0.62
McDonald’s omega	0.93	0.90	0.93	0.94	0.96	0.88

Note. ERA = Expectations Regarding Aging.

## Data Availability

Data used in this study are available upon request.

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
