# Peer review of "Acceptance of a Mobile Telepresence Robot, before Use, to Remotely Supervise Older Adults’ Adapted Physical Activity"

_ijerph, 2023, doi:10.3390/ijerph20043012_

Round 1
Reviewer 1 Report
As authors state, there could be in the later stage research after the use.
Further, often the uses develop over the time. The domestication paradigm (or other related work) could be used in the analysis how the use will settle over time. But currently the text is a feasible contribution at this stage.
Author Response
Thank you for your comments.
Comment #1. As authors state, there could be in the later stage research after the use.
Response #1. The research that tested the acceptance of the MTR after actual use is currently being submitted to another journal and is under review. The main result is that the MTR is also well accepted after use during a physical activity session adapted to older adults.
Comment #2. Further, often the uses develop over the time. The domestication paradigm (or other related work) could be used in the analysis how the use will settle over time. But currently the text is a feasible contribution at this stage.
Response #2. Thank you for your suggestion. We have added the following sentence (highlighted in yellow) in the Direction for future studies section:
"A future study using pre-test and post-test measures of acceptance might be relevant to investigate how older adults’ acceptance of the MTR would evolve after a physical activity session supervised by an APA teacher through the MTR. To go one step further, it may also be relevant to examine how MTR would be adapted to everyday life and how it would contribute to changes in everyday life, based on the domestication paradigm (Mackay & Gillespie, 1992)."We have also added the following reference:
Mackay, H., & Gillespie, G. (1992). Extending the social shaping of technology approach: Ideology and appropriation. Social Studies of Science, 22, 685-716. https://doi.org/10.1177/030631292022004006
Reviewer 2 Report
First of all, I would like to thank the editor for offering me the opportunity to review this work.
I consider the work suitable for publication. The topic is of interest within the field of geriatrics with the intention of preventing the adverse effects of aging at an early age of 55 years and older. Bringing physical exercise closer to the elderly and including it in the activities of daily living in their homes can be interesting for this group.
Through this work, the researchers have some guidelines on what this group expects from including physical exercise at home.
Given that in general, lack of adherence is common in studies with older people, it would be interesting and I encourage researchers to carry out this project and conduct the same questionnaires at the end of satisfaction.
As points of improvement for the article and for its suitability for publication I suggest:
Revise the bibliography both in the text and at the end and unify formats. In the text, for example, there are different ways of citing bibliographic references, as in lines 70, 163, 176, 193, 197, 197, 210, 210, 217, 290, 296, 334 and 347 (in some the author's name and year are mentioned and in others the author's name "et al." and year).
Author Response
Comment #1. First of all, I would like to thank the editor for offering me the opportunity to review this work. I consider the work suitable for publication. The topic is of interest within the field of geriatrics with the intention of preventing the adverse effects of aging at an early age of 55 years and older. Bringing physical exercise closer to the elderly and including it in the activities of daily living in their homes can be interesting for this group. Through this work, the researchers have some guidelines on what this group expects from including physical exercise at home.
Response #1. Thank you very much for your comments.
Comment #2. Given that in general, lack of adherence is common in studies with older people, it would be interesting and I encourage researchers to carry out this project and conduct the same questionnaires at the end of satisfaction.
Response #2. The research that tested the acceptance of the MTR after actual use is currently being submitted to another journal and is under review. The main result is that the MTR is also well accepted after use during a physical activity session with older adults.
Comment #3. As points of improvement for the article and for its suitability for publication I suggest: Revise the bibliography both in the text and at the end and unify formats. In the text, for example, there are different ways of citing bibliographic references, as in lines 70, 163, 176, 193, 197, 197, 210, 210, 217, 290, 296, 334 and 347 (in some the author's name and year are mentioned and in others the author's name "et al." and year).
Response #3. The whole article has been checked and errors are now corrected. However, when the article was signed by only two authors, the bibliographic standards of the journal require us to indicate both names.
Reviewer 3 Report
I read with interest the study presented by Mascret and Temprado, in which they describe the acceptance of a mobile telepresence robot among older subjects.
The study scope is novel, and addresses and an uncovered aspect of literature, with potential to contribute to scientific understanding.
However, few issues need to be addressed:
- The title needs to be adjusted to be reflective of the study content. Physical activity is a general term and could include day-to-day activity like going to the bathroom or walking to the kitchen for a glass of water.
- The subheading "Data analysis" should be replaced with " Statistical Analysis".
- The method of comparing the different variables is inappropriate and not fit for that purpose. If you are comparing a single variable between 2 different groups, student t-test is a correct method, provided the data is parametric. To compare more than 2 groups, provided the data is parametric, One-way ANOVA should be used. If the data is nonparametric, different methods should be used.
- It is better and more commonly used to start the Results section by describing your study cohort (participants) characteristics (age, gender, level of education, residence type and/or location, the presence of cohabitants..etc)>
- Figures are not clearly labelled. A figure title and legend (concise explanaition) should be provided for each figure.
- Figure 3 is confusing and difficult to read. Levels of significance in difference should be clarified, e.g., Perceived usefulness has * (but it is not clear when it was compared to which group).
- Figure 3 is most likely an invalid comparison as scores of entirely different components are compared, e.g., comparing oranges to eggplants.
Author Response
Comment #1. I read with interest the study presented by Mascret and Temprado, in which they describe the acceptance of a mobile telepresence robot among older subjects. The study scope is novel, and addresses and an uncovered aspect of literature, with potential to contribute to scientific understanding.
Response #1. Thank you very much for your comments.
However, few issues need to be addressed:
Comment #2. The title needs to be adjusted to be reflective of the study content. Physical activity is a general term and could include day-to-day activity like going to the bathroom or walking to the kitchen for a glass of water.
Response #2. You are right. We added "adapted physical activity" in the title to clearly identify that this is a specific type of supervised physical activity, building on the term commonly used in this field of intervention.
Comment #3. The subheading "Data analysis" should be replaced with " Statistical Analysis".
Response #3. The change has been made.
Comment #4. The method of comparing the different variables is inappropriate and not fit for that purpose. If you are comparing a single variable between 2 different groups, student t-test is a correct method, provided the data is parametric. To compare more than 2 groups, provided the data is parametric, One-way ANOVA should be used. If the data is nonparametric, different methods should be used.
Response #4. We completely agree with your point about the type of statistics to use when comparing two or three groups. However, in the present study, we are not comparing two groups with each other or three groups with each other, but we are comparing for each of the TAM variables, in succession, whether all participants have a significantly higher or lower Likert scale score than the mean. For this reason we used the "one-sample t-test" which is the most appropriate type of statistical analysis for this analysis (especially since the distribution of the data follows the normal distribution, please see Table 1).
Comment #5. It is better and more commonly used to start the Results section by describing your study cohort (participants) characteristics (age, gender, level of education, residence type and/or location, the presence of cohabitants..etc)
Response #5. We have moved the information you suggested at the beginning of the Results section (please see chapter 3.1 in the revised article).
Comment #6. Figures are not clearly labelled. A figure title and legend (concise explanation) should be provided for each figure.
Response #6. We completely understand your comment. However, IJERPH's editorial standards require us to offer only one title to the figure.
Comment #7. Figure 3 is confusing and difficult to read. Levels of significance in difference should be clarified, e.g., Perceived usefulness has * (but it is not clear when it was compared to which group).Comment #8. Figure 3 is most likely an invalid comparison as scores of entirely different components are compared, e.g., comparing oranges to eggplants.
Responses #7 and #8. Our response to your comments #7 and #8 is related to our response to your comment #4. The level of significance corresponds to the comparison of the scores of each variable with the mean of the scale (i.e., 3) for the whole sample but it is not a comparison between groups. However, we have added clarification in the Statistical Analysis section to avoid confusion on this issue.